# A Longitudinal Study of Exposure to Manganese and Incidence of Metabolic Syndrome

**DOI:** 10.3390/nu14204271

**Published:** 2022-10-13

**Authors:** Emily Riseberg, Kenneth Chui, Katherine A. James, Rachel Melamed, Tanya L. Alderete, Laura Corlin

**Affiliations:** 1Department of Public Health and Community Medicine, Tufts University School of Medicine, Boston, MA 02111, USA; 2Department of Nutrition, Harvard T.H. Chan School of Public Health, Boston, MA 02115, USA; 3Department of Environmental and Occupational Health, Colorado School of Public Health, University of Colorado, Anschutz Medical Campus, Aurora, CO 80045, USA; 4Department of Biological Sciences, University of Massachusetts, Lowell, MA 01854, USA; 5Department of Integrative Physiology, University of Colorado, Boulder, CO 80309, USA; 6Department of Civil and Environmental Engineering, Tufts University School of Engineering, Medford, MA 02155, USA

**Keywords:** manganese, metabolic syndrome, rural health, urinary metals, longitudinal, Bayesian kernel machine regression

## Abstract

The association between manganese (Mn) and metabolic syndrome (MetS) is unclear, and no prior study has studied this association longitudinally. The aim of this study was to assess longitudinal associations of Mn exposure with MetS and metabolic outcomes. We used data from the San Luis Valley Diabetes Study (SLVDS), a prospective cohort from rural Colorado with data collected from 1984–1998 (n = 1478). Urinary Mn was measured at baseline (range = 0.20–42.5 µg/L). We assessed the shape of the cross-sectional association between Mn and MetS accounting for effect modification by other metals at baseline using Bayesian kernel machine regression. We assessed longitudinal associations between baseline quartiles of Mn and incident MetS using Fine and Gray competing risks regression models (competing risk = mortality) and between quartiles of Mn and metabolic outcomes using linear mixed effects models. We did not observe evidence that quartiles of Mn were associated with incident MetS (*p*-value for trend = 0.52). Quartiles of Mn were significantly associated with lower fasting glucose (*p*-value for trend < 0.01). Lead was found to be a possible effect modifier of the association between Mn and incident MetS. Mn was associated with lower fasting glucose in this rural population. Our results support a possible beneficial effect of Mn on diabetic markers.

## 1. Introduction

The prevalence of metabolic syndrome (MetS) has been increasing across all sociodemographic groups such that by 2015–2016, approximately 35% of adults in the United States (US) had MetS [1]. MetS represents the co-occurrence of multiple cardiometabolic risk factors [2], generally including at least some sub-set of the following five: obesity, low values of high-density lipoproteins (HDL), high triglycerides, hyperglycemia, and hypertension [2,3]. The presence of MetS is more predictive of cardiovascular disease risk and diabetes risk than any single one of the five factors alone [3]. MetS is also associated with other outcomes, such as nonalcoholic fatty liver disease, chronic kidney disease, neurodegenerative disorders, and some cancers [3,4]. Having a lower level of education, older age, less physical activity, higher meat intake, and residence in rural regions are each associated with increased likelihood of MetS [4,5,6]. Additionally, exposure to some heavy metals and metalloids (hereafter referred to as metals), such as arsenic, selenium, and zinc, has been associated with higher MetS prevalence cross-sectionally, whereas exposure to other metals (e.g., lead) has been found to be inversely associated with MetS [7,8]. Our understanding of the relationships between metal exposure and MetS is limited by a dearth of longitudinal analyses.

Manganese (Mn) is an essential metal found naturally in bedrock and soil. People are typically exposed to Mn through food, water, and air [9]. Mn naturally exists in most foods, although certain shellfish, grains, beans, nuts, and tea have higher amounts of Mn [9,10]. Sources of Mn in ambient air include mining, automobile exhaust, and industrial processes involving Mn-containing products (such as steel production) [9]. Mn in small amounts is essential for health, but populations that are exposed to high levels of Mn have been found to be at higher risk of adverse nervous system and respiratory effects [9,10,11]. The recommended daily intake of Mn from dietary sources is 2.3 mg per day in male adults and 1.8 mg per day in female adults [12]. Excess levels of Mn in the body may accumulate when the excretion system is impaired or undeveloped, when the Mn transporter malfunctions, or if there is excess environmental exposure to Mn (e.g., as may occur for people in occupations such as mining and steel making) [10,13]. Both low and high levels of Mn may be associated with increased likelihood of MetS because both Mn deficiency and excess Mn are associated with increased oxidative stress and mitochondrial dysfunction [13].

All prior epidemiological investigations of which we are aware that considered putative associations between Mn exposure and MetS used cross-sectional (12 studies [7,14,15,16,17,18,19,20,21,22,23,24]) or case–control designs (3 studies [25,26,27,28]). Overall, these studies have not observed significant associations between Mn measured in diet [14,20,25], serum [15,22,27], whole blood [7,16,18,21], plasma [23,24], or urine [16,17,19,26] with MetS. Similarly, a meta-analysis of many of these studies did not find significant associations when pooling results by measurement method [28]. However, results have been somewhat inconsistent, especially with regards to differences by sex [14,16,19,20,22]. For example, one cross-sectional study among adults in China found that self-reported dietary Mn intake was associated with an increased likelihood of having MetS in women and a decreased likelihood in men [20]. This study also observed associations between dietary Mn intake and MetS components (e.g., low values of HDL cholesterol for both women and men; interactions by sex for abdominal obesity) [20]. In contrast, bivariate analyses from a Korean cross-sectional study suggested that women with MetS had lower dietary Mn intake than women without MetS (though these bivariate associations were not observed among men, and dietary intake was not associated with MetS among either women or men after adjusting for covariates) [14]. Additionally, previous studies have been inconsistent with respect to how they handled potential non-linear relationships between Mn exposure and MetS. For example, some studies only considered linear or bivariate relationships [19,26], some considered associations by quantiles of exposure [7,14,15,18,20,23,24,25,27], and some used analytic strategies that did not presume a functional form [16,21,22]. Of the studies that considered the potential for U-shaped associations between Mn and MetS, both provided evidence for such a non-linear relationship [16,21]. Finally, of the previous 15 studies, only seven accounted for co-exposure to other metals [7,15,16,17,20,21,27]. Of these, all adjusted for other metals in the models [7,15,16,17,20,21,27], and some employed Bayesian kernel machine regression to account for effect modification by other metals [16,17]. One study used principal components analysis to consider patterns of metal exposure and detected a methylmercury-manganese pattern that was not found to be significantly associated with MetS [7].

To further our understanding of the relationships between Mn and MetS, our objectives were to: (1) assess cross-sectional associations between measured Mn and MetS accounting for potential non-linear relationships and effect modification by metal co-exposures; (2) assess longitudinal associations between measured Mn and MetS (as well as MetS components); and (3) assess whether sex modified the associations between Mn and MetS.

## 2. Materials and Methods

### 2.1. Study Population

The San Luis Valley Diabetes Study (SLVDS) is a prospective cohort study designed to assess risk factors for type 2 diabetes in a rural population in Colorado. The San Luis Valley covers approximately six counties in Colorado, including Alamosa and Conejos counties, and is primarily bi-ethnic [29]. The details of the SLVDS have been described elsewhere [30,31]. Briefly, participants had to be: (1) 20–74 years of age, (2) current residents of either Alamosa or Conejos counties, (3) able to complete the interview in either English or Spanish, and (4) mentally competent. Individuals with diabetes at baseline were recruited through advertisements and medical records, and individuals without diabetes at baseline were recruited through a two-stage geographic sampling procedure. This two-stage procedure first incorporated maps, directories, and other geographic information to sample about one-fifth of the households in the two counties. Then, individuals were recruited within age, sex, ethnic, and county strata to reflect the demographic makeup of diabetes in the study region. Baseline data collection occurred from 1984 to 1988, and follow-up data collection occurred over the next 10 years. Data concerning demographic, clinical, and behavioral characteristics as well as diagnoses of cardiometabolic outcomes were collected at each visit. Data on mortality and vital statistics were collected between visits through record searches and interviews over the phone [32].

### 2.2. Urinary Metal Assessment

Urine samples of approximately 120 mL were collected at baseline in trace-free metal containers and aliquoted in 5 mL tubes. These samples were first stored in a freezer at −80 °C and then transferred to the Colorado State Department of Public Health and Environment chemistry laboratory in 2003 where they were stored at −80 °C until laboratory analysis in 2008 and 2015 by the Colorado Department of Public Health and Environment and the Columbia University Metals Laboratory, respectively. The laboratory methods at both locations met the standards of the Environmental Protection Agency and Clinical Laboratory Improvement Amendment. The storage procedure for these samples has been shown to be reliable, as thawing and refreezing does not compromise the sample [33]. The sample was thawed and mixed, and a <1 mL aliquot was used for analysis. An inductively coupled argon plasma instrument with a mass spectrometer was used to detect the metal concentrations (µg/L) of arsenic (As), barium (Ba), cadmium (Cd), cobalt (Co), chromium (Cr), cesium (Cs), copper (Cu), Mn, molybdenum (Mo), lead (Pb), selenium (Se), thallium (Tl), tungsten (W), and zinc (Zn; detection limit of all metals = 1 part in 10) [31,34,35]. Of the 1478 participants who were eligible for this analysis, 242 (16%) had a Mn concentration below the level of detection. Concentrations below the level of detection were assigned the square root of the detection limit divided by two.

### 2.3. Metabolic Syndrome Definition and Measurement

We used a modified version of the National Cholesterol Education Program/Adult Treatment Panel (NCEP/ATP) III guidelines adapted to the SLVDS dataset [3]. Participants were considered to have MetS if they had three or more of the following outcomes: (1) waist–hip ratio (waist circumference divided by iliac circumference) >0.90 for males, waist–hip ratio >0.85 for females, or a body mass index (BMI; measured weight divided by measured height squared) >30 kg/m^2^, (2) HDL <40 mg/dL for males or <50 mg/dL for females, (3) triglycerides ≥150 mg/dL, (4) fasting glucose ≥100 mg/dL or diabetes (includes people who self-reported being diagnosed with diabetes via an oral glucose tolerance test or prescribed insulin or oral hyperglycemic medication and later confirmed through medical records), or (5) measured systolic blood pressure ≥130 mmHg or diastolic blood pressure ≥85 mmHg or self-reported current use of antihypertensive medications. All components except those measuring obesity were from the NCEP/ATP III guidelines. For obesity, we used the World Health Organization 1999 definition since waist–hip ratio is a stronger predictor of obesity than waist circumference in older study populations such as the SLVDS [3,36]. Blood samples were analyzed according to established protocols: HDL and triglyceride measurements were obtained from the blood collection using enzymatic methods [32]. Glucose values were obtained from those who fasted at least eight hours using a glucose tolerance test with the Chemstrip bG (Boehringer-Mannheim Diagnostics, Indianapolis, IN) and measured using the glucose oxidase method [30,37]. Blood pressure was measured three times, and the average of the second and third was used as the measurement.

### 2.4. Covariate Measurement

Covariates were identified using an evidence-based directed acyclic graph and included: sex (female/male), age (years; continuous), ethnicity (Hispanic/non-Hispanic), annual gross household income (<USD 10,000, USD 10,000-24,999, ≥USD 25,000), smoking status (<100 cigarettes in lifetime (never), ≥100 cigarettes in lifetime and does not currently smoke (former), ≥100 cigarettes in lifetime and currently smokes (current), caloric intake (kcal/day), and urinary creatinine (g/L) [38]. Caloric intake (kcal/day) was obtained from a food frequency questionnaire [39]. Urinary creatinine (g/L) was determined using a colorimetric assay with the Jaffe reaction [40].

### 2.5. Statistical Analysis

Of the total of 1823 SLVDS participants, individuals were excluded from our analyses if they had complications with the urine sample (*n* = 28), were missing baseline data on Mn (*n* = 186), were missing baseline data on covariates (*n* = 130), or had implausible follow-up time (*n* = 1). No participants were missing data on MetS at baseline (defined as missing three or more component values), and 609 participants who did not have MetS at baseline had data for at least one follow-up study visit. One additional participant was excluded from longitudinal analyses for having two study follow-up visits within 30 days of each other. Thus, the sample size was 1478 participants for the cross-sectional analysis and 608 participants for the longitudinal MetS analysis. For the secondary analyses examining associations with MetS components, the sample size varied from 1475 to 1477 participants, depending on missing data for the specific outcome.

We compared baseline covariates among individuals with Mn values above and below the median using *t* tests for continuous variables and chi-squared tests for categorical variables. We examined correlations among metals using Spearman’s rank correlation coefficients. To evaluate potential non-linear cross-sectional associations with MetS and effect modification by other metals (i.e., As, Ba, Cd, Co, Cr, Cs, Cu, Mo, Pb, Se, Tl, W, and Zn) at baseline, we used Bayesian kernel machine regression (BKMR) [41]. Due to the skewed distribution of the metals, the metal values in the BKMR models were natural log-transformed. The cross-sectional association between Mn and MetS appeared nonlinear, so further analyses used quartiles of Mn rather than assessing it as a continuous measure.

We assessed longitudinal associations between quartiles of Mn and MetS using Fine and Gray competing risks regression (competing event = mortality). We assessed longitudinal associations between quartiles of Mn and individual MetS component outcomes (i.e., waist–hip ratio, BMI, total HDL, triglycerides, fasting glucose, systolic blood pressure, and diastolic blood pressure) using linear mixed effects models with a random intercept for each participant. In all longitudinal analyses, three sets of models were fit: (1) models adjusted only for urinary creatinine (crude models); (2) models adjusted for baseline values of all covariates identified a priori (primary models); and (3) sex-stratified models adjusted for all covariates in the primary models except for sex. Candidate metals that acted as effect modifiers in the BKMR models were identified using a liberal threshold. For each of these metals, we tested for effect modification of Mn by adding quartile values of the metal to the adjusted Fine and Gray models as well as an interaction term between each of the other metals and quartiles of Mn to see if the sub-distribution hazard ratio differed from the model without the interaction term (separate models for each potential effect modifier). We conducted a sensitivity analysis for the longitudinal models that excluded participants who had Mn levels that were below the limit of detection.

In all analyses, associations were considered statistically significant if *p* < 0.05. All analyses were conducted with R version 4.1.0 (R Core Team, Vienna, Austria). We used the bkmr (version 0.2.0), cmprsk (version 2.2-11), lme4 (version 1.1-27.1), lmerTest (version 3.1-3), corrplot (version 0.92), and ggplot2 (version 3.3.5) packages in R [41,42,43,44,45,46,47].

## 3. Results

### 3.1. Baseline Characteristics

At baseline, the median (25th, 75th percentiles) Mn value was 0.63 (0.33, 1.13) µg/L. Mn values ranged from 0.20 to 42.5 µg/L, and the geometric mean (95% confidence interval (CI)) was 0.67 (0.64, 0.70) µg/L. The medians, 25th, and 75th percentiles for the other metals are presented in Appendix A, and the correlations among metals are shown in Figure 1. Mn was most highly correlated with Cu (ρ = 0.66), Tl (ρ = 0.43), and Se (ρ = 0.41).

Other baseline characteristics of the study sample are presented in Table 1. Briefly, 51% of the participants were female, 47% were Hispanic, and the median (25th, 75th percentiles) age was 55 (45, 64) years old. People with higher Mn at baseline were significantly more likely to be male (*p* = 0.04), older (*p* = 0.01), Hispanic (*p* = 0.02), have obesity (*p* = 0.01), and not have high fasting glucose (*p* = 0.02). The likelihood of having high Mn at baseline also differed by smoking status (*p* < 0.01). Neither total household income nor total caloric intake was found to differ significantly by Mn status (Table 1). At baseline, 56% of the sample had MetS, and 88% had obesity, the largest MetS component in this population (Table 1).

### 3.2. Outcome Follow-up

The risk of MetS among the 608 participants who did not have MetS at baseline was 37% over a mean (standard deviation) of 7 (4) years of follow-up (median number of study visits = 4; mean time between study visits = 3 years; minimum time between visits = 37 days; maximum time between visits = 13 years). Among the 608 participants with follow-up data, 107 died (18%) without developing MetS. Compared to the population in the cross-sectional analyses (*n* = 1478) that included participants with MetS at baseline, participants in the longitudinal analyses were, on average, younger at baseline (mean age = 51 vs. 55 years), more likely to be female (58% vs. 51%), and more likely to be non-Hispanic (61% vs. 53%).

### 3.3. Cross-sectional Analyses

Accounting for the possibility of non-linear associations, we did not observe strong evidence that natural log-transformed Mn was cross-sectionally associated with MetS, although the relationship appeared to be somewhat non-linear (Appendix A). Natural log-transformed values of Ba, Cd, Co, Cs, Mo, Pb, and Zn appeared to be potential effect modifiers (Figure 2). The other metals did not appear to be effect modifiers (results not shown).

### 3.4. Longitudinal Analyses

In longitudinal analyses, we did not observe evidence that quartiles of Mn were associated with incident MetS in the crude model (*p*-value for trend = 0.41) or the adjusted model (*p*-value for trend = 0.52). These associations were not modified by sex (Table 2). Quartiles of Mn were significantly associated with lower fasting glucose levels comparing highest quartile of Mn to lowest (β = −12.6; 95% CI= −20.3, −4.9; *p*-value for trend < 0.01). This association was found in males (*p*-value for trend < 0.01) and females (*p*-value for trend = 0.048; Table 3). In crude models but not adjusted models, higher quartiles of Mn were associated with lower waist–hip ratio overall, with lower triglycerides in males, and with higher BMI and systolic blood pressure in females (Appendix A).

In sensitivity analyses excluding participants with Mn levels below the limit of detection (*n* = 242 removed), results remained similar. Mn was not associated with MetS longitudinally (adjusted *p*-value for trend = 0.64 overall, 0.35 for males, and 0.74 for females). Mn was significantly associated with lower fasting glucose overall and among males, and with lower triglycerides among males (Appendix A).

In separate longitudinal models accounting for effect modification by other metals, associations between quartiles of Mn and MetS remained similar (Table 4). The model with an interaction term between quartiles of Mn and quartiles of Pb yielded a significant association between Mn and MetS comparing the highest quartile of Mn to the lowest (sub-distribution hazard ratio = 3.36; 95% CI = 1.11, 10.17; *p*-value for trend = 0.01). None of the interaction terms between Mn and other metals were significant except for interaction terms between certain quartiles of Pb and Mn.

## 4. Discussion

In our longitudinal study of adults in rural Colorado, we did not observe significant associations between urinary Mn and incident MetS. This finding was consistent in sex-stratified models, when adjusting for co-exposure to other metals, and when accounting for potential non-linear relationships between Mn and MetS. Our study was the first longitudinal cohort study that has examined the association between Mn and incident MetS. Our overall null findings were consistent with previous cross-sectional and case–control studies [7,14,15,17,18,19,21,22,23,24,26]. Additionally, we observed that Mn was significantly and inversely associated with fasting glucose, overall and in sex-stratified models. Our study adds to the growing body of evidence that Mn is not associated with adverse metabolic outcomes.

This longitudinal study differed methodologically from the three previous cross-sectional analyses of the associations between Mn and MetS in the US in that we focused on a rural population that experiences more population-level stress and health burden [7,16,21,48]. For example, the Mn concentrations were somewhat higher in our study sample than in the general US population during the same time period (SLVDS geometric mean (95% CI) = 0.67 (0.64, 0.70) µg/L; US National Health and Nutrition Examination Survey III (1988–1994) geometric mean (95% CI) Mn level = 0.53 (0.46, 0.61) µg/L) [49]. The difference in Mn exposure likely was not due to air exposure in occupational settings, such as factories that use metals [9]. Instead, the differences in Mn exposure were likely due to Mn in the diet, soil, and water. To determine the extent to which participants were being exposed through food, we would need other measures that estimate intake of individual food items high in manganese rather than total calories per day. While dietary Mn intake was not calculated in the SLVDS, we encourage others with access to dietary Mn data and a biomarker for Mn (e.g., urine, blood) to investigate the extent to which the two correlate. Additionally, although Mn supplementation is not common because it is found in many foods, some individuals who have low dietary intake opt to take Mn supplements and it may, therefore, be helpful to include information on Mn supplementation dosage and duration to more accurately capture total Mn exposure [9].

The finding that Mn was inversely associated with fasting glucose is consistent with some, but not all, of the previous studies [14,16,24,50]. Even in studies where inverse associations were observed, there has been evidence for non-linear relationships (e.g., in one cross-sectional study, Mn was inversely associated with fasting glucose among men comparing only the second and third quartiles of Mn exposure, but not the fourth quartile, to the first quartile) [16]. In our study, it is possible that the observed inverse association with fasting glucose was attributable to healthier dietary patterns among people with higher Mn concentrations. Healthy plant-based foods such as whole grains, nuts, and beans are sources of Mn [51], and these foods along with a healthy plant-based diet have been associated with improved markers of diabetes [52,53,54,55]. However, because Mn is also present in a number of unhealthy foods [51], and Mn was not significantly associated with total caloric intake in baseline bivariate analyses, it is likely that higher consumption of Mn-rich foods rather than an overall healthy diet drove the association between Mn and fasting glucose. Shellfish, while a major source of dietary Mn in general [9,51], is not a common dietary source for this rural cohort. Future studies could further investigate associations with Mn-rich foods and potential confounding by overall diet, particularly among populations that consume more shellfish than did participants in the SLVDS.

Although we did not observe an association between Mn and incident MetS, even accounting for a potential non-linear relationship, there are several mechanisms through which both inadequate and excess Mn could theoretically increase risk of MetS and metabolic outcomes. For example, Mn metalloenzymes, such as Mn superoxide dismutase (MnSOD), help to reduce mitochondrial oxidative distress [13]. However, excess Mn intake can lead to overproduction of reactive oxygen species (ROS), contributing to MetS development through insulin resistance and increased blood pressure [13,56]. Pb, like Mn, also produces ROS and can lead to oxidative damage when consumed in toxic amounts [57], which could explain why Pb modified the association between Mn and MetS. Furthermore, Mn is required for biological processes including carbohydrate and lipid metabolism, and these can contribute to dysregulated MetS components when impaired [13,58,59]. Specifically, Mn deficiency can impair glucose transport, which could explain the inverse association we observed with fasting glucose [60]. Sex differences in the unadjusted associations between Mn and metabolic outcomes could be attributable to differences in hormone levels, as sex hormones contribute to diabetes and MetS differentially in females and males [61]. Given the integral role of glucose homeostasis in several potential mechanisms and the significant inverse findings with fasting glucose we observed, previous significant associations observed between Mn and MetS in a minority of previously published studies may have been primarily driven through this pathway [16,20,25,27].

Despite the general agreement of our results with previous studies that assessed associations between Mn and MetS using a variety of Mn assessment methods, our choice of biological exposure matrix (urine) may have complicated the interpretation of the results. Little Mn is excreted in the urine (7 nmol per g of creatinine for healthy, non-smoking males and 9 nmol per day for healthy, non-smoking females), and the excretion amount may not be substantially affected by increased Mn intake [9,62]. Mn measurements in urine tend to more accurately predict Mn deficiency than excess intake [12]. That said, we had sufficient exposure contrast with urinary Mn measurement concentrations from below 1 to 42.5 µg/L. However, another potential limitation of urinary Mn is the exposure window to which it likely corresponds. Whereas Mn measured in blood may indicate body burden, urinary Mn may be a better indicator of recent exposure and thus may not be as useful as a biological matrix for longitudinal studies [9]. Nevertheless, in one cross-sectional study that separately analyzed urinary Mn-MetS and blood Mn-MetS associations, the blood Mn-MetS results were somewhat more consistent with our generally null findings than were the urinary Mn-MetS results [16]. Similarly, and analogously to our longitudinal assessment of sex-specific urinary Mn-fasting glucose associations, another cross-sectional study found that both blood and urinary Mn had sex-specific associations with diabetes markers (e.g., fasting glucose) [50]. Furthermore, urinary Mn has an important advantage over Mn assessed through the use of dietary intake data (as has been done in multiple studies) [14,20,25], since it can integrate across different external sources of exposure and can limit differential exposure misclassification. While limited compared to dietary assessment, there is still some potential for differential exposure misclassification with urinary Mn measures, especially if the large percentage of the sample with Mn values below the limit of detection (16% of measurements were below this threshold) were low due to factors related to incident MetS. Thus, given the relative strengths and weaknesses of the different exposure methods, we suggest that future longitudinal studies consider the use of blood Mn.

Additionally, our study had several limitations. Our results may not generalize to populations with different exposure distributions of Mn. This could be relevant, for example, if regulations are passed to limit Mn exposure in certain settings. Relatedly, since we used data from a bi-ethnic rural cohort, we do not know if our results would be generalizable to other settings (such as urban locations) or in populations that are more diverse. We were also unable to assess associations due to specific routes of exposure (or sources of exposure) to Mn in this study population. Finally, we may have observed some significant associations just due to chance given the large number of comparisons.

Our study also has several strengths. First, we were able to conduct the first known longitudinal assessment of the association between Mn and MetS. Additionally, we accounted for mortality as a competing risk—which may have been particularly relevant as 18% of the sample included in the longitudinal analysis died without having developed MetS over an average of seven years. Finally, we were able to adjust for covariates selected based on an evidenced-based process [38], account for the possibility of non-linear relationships, and consider the potential role of co-exposure to other metals in a longitudinal setting.

## 5. Conclusions

Urinary Mn was not associated with incident MetS in this rural Colorado cohort; however, Mn was inversely associated with fasting glucose in the longitudinal analysis, indicating that Mn consumption could help reduce risk of diabetes. Future studies should consider alternative methods of exposure assessment.

## Figures and Tables

**Figure 1 nutrients-14-04271-f001:**
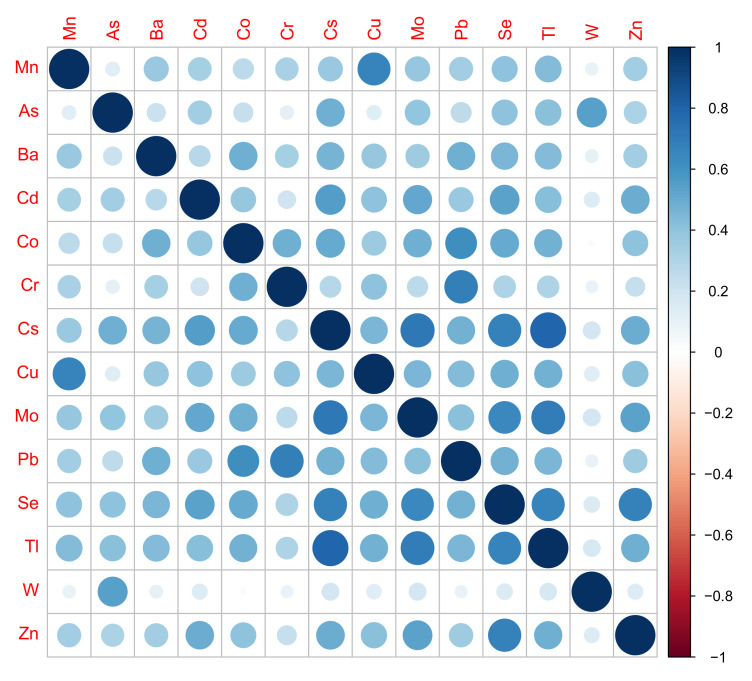
Spearman rank correlations among metals included in this analysis (*n* = 1478). Larger circles indicate larger magnitude of correlations. Color of circles indicates magnitude and direction of correlation.

**Figure 2 nutrients-14-04271-f002:**
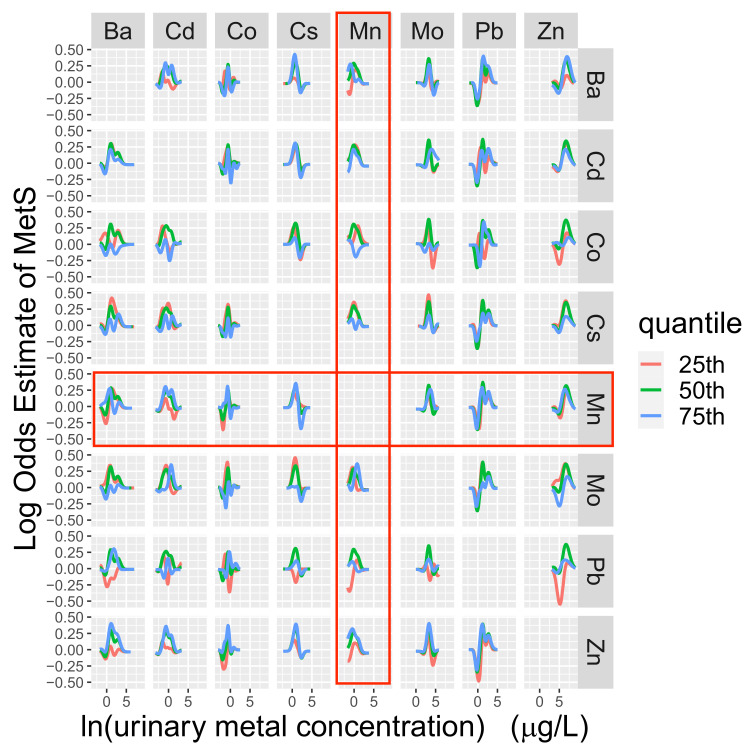
Bayesian kernel machine regression model of the cross-sectional associations between natural log-transformed (ln) metals and metabolic syndrome (MetS). Figure Legend: Metals along the y axis are categorized into quartiles. Red outlined plots represent interactions between Mn and other metals. Parallel dose–response lines indicate that interaction is not present. Models were adjusted for baseline values of sex, age (years), ethnicity (Hispanic, non-Hispanic), annual gross household income (<USD 10,000, USD 10,000–24,999, ≥USD 25,000), smoking status (<100 cigarettes in lifetime (never), ≥100 cigarettes in lifetime and does not currently smoke (former), ≥100 cigarettes in lifetime and currently smokes (current)), caloric intake (kcal/day), and urinary creatinine (g/L). Metabolic syndrome was defined as having three or more of the following outcomes: (1) waist–hip ratio (waist circumference divided by iliac circumference) >0.90 for males, waist–hip ratio >0.85 for females, or a body mass index (measured weight divided by measured height squared) >30 kg/m^2^, (2) high density lipoprotein <40 mg/dL for males or <50 mg/dL for females, (3) triglycerides ≥150 mg/dL, (4) fasting glucose ≥100 mg/dL or diabetes (includes people who self-reported being diagnosed with diabetes via an oral glucose tolerance test or prescribed insulin or oral hyperglycemic medication), or (5) measured systolic blood pressure ≥130 mmHg or diastolic blood pressure ≥85 mmHg or self-reported current use of antihypertensive medications.

**Table 1 nutrients-14-04271-t001:** Baseline characteristics ^a^ of the sample contrasted by the median Mn value.

	All Participants	Mn ≤ 0.63	Mn > 0.63	*p-Value* ^b^
Total	1478 (100.0)	743 (50.0)	735 (50.0)	-
Sex				0.04 *
Males	723 (48.9)	344 (46.3)	379 (51.6)	
Females	755 (51.1)	399 (53.7)	356 (48.4)	
Age (years)	55.1 (45.3, 63.6)	54.2 (44.5, 62.5)	56.5 (45.9, 64.1)	0.01 *
Ethnicity				0.02 *
Hispanic	696 (47.1)	328 (44.1)	368 (50.1)	
Non-Hispanic	782 (52.9)	415 (55.9)	367 (49.9)	
Total Household Income				0.84
<USD 10,000	442 (29.9)	217 (29.2)	225 (30.6)	
USD 10,000–24,999	541 (36.6)	274 (36.9)	267 (36.3)	
≥USD 25,000	495 (33.5)	252 (33.9)	243 (33.1)	
Smoking Status				<0.01 *
Never (<100 cigarettes in lifetime)	645 (43.6)	251 (33.8)	394 (53.6)	
Current (≥100 cigarettes and currently smokes)	361 (24.4)	216 (29.1)	145 (19.7)	
Former (≥100 cigarettes and currently does not smoke)	472 (31.9)	276 (37.1)	196 (26.7)	
Caloric intake (kcal/day)	1463 (1114, 1838)	1448 (1108, 1829)	1472 (1130, 1847)	0.76
Metabolic syndrome prevalence ^c^	831 (56.2)	414 (55.7)	417 (56.7)	0.69
Obesity	1307 (88.4)	640 (86.1)	667 (90.7)	0.01 *
Low values of high-density lipoprotein	645 (43.6)	306 (41.2)	339 (46.1)	0.06
High triglycerides	715 (48.4)	359 (48.3)	356 (48.4)	0.96
High fasting glucose	738 (49.9)	393 (52.9)	345 (46.9)	0.02 *
High blood pressure	648 (43.8)	333 (44.8)	315 (42.9)	0.45

* *p* < 0.05. ^a^ Values are median (25th, 75th percentiles) or *n* (%). ^b^
*p*-values generated from chi-squared (categorical) and *t* tests (continuous) comparing baseline covariates by Mn baseline value being at or below the median (0.63 µg/L) versus above the median. ^c^ Metabolic syndrome was defined as having three or more of the following outcomes: (1) waist–hip ratio (waist circumference divided by iliac circumference) >0.90 for males, waist–hip ratio >0.85 for females, or a body mass index (BMI; measured weight divided by measured height squared) >30 kg/m^2^, (2) high density lipoprotein (HDL) <40 mg/dL for males or <50 mg/dL for females, (3) triglycerides ≥150 mg/dL, (4) fasting glucose ≥100 mg/dL or diabetes (includes people who self-reported being diagnosed with diabetes via an oral glucose tolerance test or prescribed insulin or oral hyperglycemic medication), or (5) measured systolic blood pressure ≥130 mmHg or diastolic blood pressure ≥85 mmHg or self-reported current use of antihypertensive medications.

**Table 2 nutrients-14-04271-t002:** Sub-distribution hazard ratios and 95% confidence intervals for the longitudinal associations ^a^ between quartiles of manganese and metabolic syndrome (*n* = 608) ^b^.

	Quartile 2 0.33–0.63 µg/L	Quartile 3 0.63–1.13 µg/L	Quartile 4 1.13–42.5 µg/L	*p-Value for T-rend*
Crude model ^c^ (*n* = 608)	1.39 (0.95, 2.02)	1.11 (0.74, 1.67)	1.30 (0.88, 1.92)	0.41
Males (*n* = 254)	1.45 (0.80, 2.63)	1.00 (0.56, 1.79)	1.21 (0.71, 2.07)	0.76
Females (*n* = 354)	1.39 (0.85, 2.30)	1.20 (0.69, 2.10)	1.39 (0.80, 2.42)	0.41
Adjusted model ^d^ (*n* = 608)	1.42 (0.97, 2.08)	1.11 (0.74, 1.68)	1.26 (0.84, 1.89)	0.52
Males (*n* = 254)	1.51 (0.81, 2.81)	0.97 (0.53, 1.80)	1.25 (0.70, 2.21)	0.71
Females (*n* = 354)	1.40 (0.84, 2.31)	1.20 (0.68, 2.10)	1.38 (0.77, 2.47)	0.43

^a^ Longitudinal analysis assessed using Fine and Gray competing risks regression with the competing event of mortality. The reference group for manganese is the first quartile (≤0.33 µg/L). ^b^ Metabolic syndrome was defined as having three or more of the following outcomes: (1) waist–hip ratio (waist circumference divided by iliac circumference) >0.90 for males, waist–hip ratio >0.85 for females, or a body mass index (measured weight divided by measured height squared) >30 kg/m^2^, (2) high density lipoprotein <40 mg/dL for males or <50 mg/dL for females, (3) triglycerides ≥150 mg/dL, (4) fasting glucose ≥100 mg/dL or diabetes (includes people who self-reported being diagnosed with diabetes via an oral glucose tolerance test or prescribed insulin or oral hyperglycemic medication), or (5) measured systolic blood pressure ≥130 mmHg or diastolic blood pressure ≥85 mmHg or self-reported current use of antihypertensive medications. ^c^ Adjusted for urinary creatinine (g/L). ^d^ Adjusted for baseline values of sex (except in sex-stratified models), age (years), ethnicity (Hispanic, non-Hispanic), annual gross household income (<USD 10,000, USD 10,000–24,999, ≥USD 25,000), smoking status (<100 cigarettes in lifetime (never), ≥100 cigarettes in lifetime and does not currently smoke (former), ≥100 cigarettes in lifetime and currently smokes (current)), caloric intake (kcal/day), and urinary creatinine (g/L).

**Table 3 nutrients-14-04271-t003:** Adjusted longitudinal associations ^a^ between baseline quartiles of manganese and metabolic outcomes.

	Quartile 2 0.33–0.63 µg/L	Quartile 3 0.63–1.13 µg/L	Quartile 4 1.13–42.5 µg/L	*p-Value for Trend*
Waist–hip ratio; *n = 1477*	0.000 (−0.006, 0.005)	−0.001 (−0.007, 0.004)	0.000 (−0.006, 0.006)	0.96
Males	0.003 (−0.004, 0.01)	0.000 (−0.007, 0.008)	0.000 (−0.007, 0.007)	0.92
Females	−0.003 (−0.011, 0.006)	−0.002 (−0.011, 0.007)	0.001 (−0.009, 0.011)	0.79
Body mass index (kg/m^2^); *n = 1477*	−0.14 (−0.82, 0.54)	−0.04 (−0.74, 0.67)	0.17 (−0.55, 0.90)	0.61
Males	0.00 (−0.84, 0.84)	−0.60 (−1.44, 0.25)	−0.16 (−1.00, 0.68)	0.51
Females	−0.10 (−1.13, 0.94)	0.49 (−0.62, 1.60)	0.62 (−0.56, 1.80)	0.21
High-density lipoprotein (mg/dL); *n = 1476*	−1.01 (−2.74, 0.73)	−1.02 (−2.83, 0.78)	−0.18 (−2.03, 1.67)	0.87
Males	−1.02 (−3.33, 1.29)	−0.01 (−2.34, 2.31)	−0.14 (−2.45, 2.16)	0.91
Females	−1.64 (−4.17, 0.90)	−2.00 (−4.71, 0.72)	−0.56 (−3.45, 2.33)	0.65
Triglycerides (mg/dL); *n = 1477*	−12.5 (−29.9, 4.9)	−10.1 (−28.1, 7.9)	−17.8 (−36.2, 0.7)	0.09
Males	−1.3 (−21.8, 19.2)	−7.2 (−27.7, 13.4)	−13.9 (−34.2, 6.5)	0.16
Females	−17.3 (−45.8, 11.2)	−13.8 (−44.3, 16.8)	−18.6 (−51.1, 14.0)	0.33
Fasting glucose (mg/dL); *n = 1475*	1.0 (−6.2, 8.2)	−9.4 (−16.9, −1.9)	−12.6 (−20.3, −4.9)	<0.01 *
Males	4.4 (−5.7, 14.6)	−14.9 (−25.1, −4.6)	−15.3 (−25.5, −5.2)	<0.01 *
Females	0.3 (−9.9, 10.4)	−4.8 (−15.7, 6.1)	−10.9 (−22.5, 0.6)	0.048 *
Systolic blood pressure (mmHg); *n = 1477*	−2.1 (−4.4, 0.2)	−0.8 (−3.2, 1.6)	−2.2 (−4.7, 0.2)	0.17
Males	−1.0 (−4.1, 2.2)	−3.1 (−6.3, 0.1)	−2.7 (−5.8, 0.5)	0.06
Females	−3.2 (−6.5, 0.2)	0.8 (−2.7, 4.4)	−2.2 (−6.0, 1.6)	0.71
Diastolic blood pressure (mmHg); *n = 1477*	−1.2 (−2.3, 0.0)	−1.3 (−2.5, −0.1)	−0.8 (−2.0, 0.4)	0.22
Males	−1.0 (−2.7, 0.7)	−2.3 (−4.0, −0.6)	−1.2 (−2.9, 0.5)	0.11
Females	−1.4 (−2.9, 0.2)	−0.4 (−2.0, 1.2)	−0.4 (−2.2, 1.3)	0.91

* *p* < 0.05. ^a^ Associations assessed using linear mixed effects models with a random intercept for each participant. The reference group for manganese is the first quartile (≤0.33 µg/L). Models are adjusted for baseline values of sex (except in sex-stratified models), age (years), ethnicity (Hispanic, non-Hispanic), annual gross household income (<USD 10,000, USD 10,000–24,999, ≥USD 25,000), smoking status (<100 cigarettes in lifetime (never), ≥100 cigarettes in lifetime and does not currently smoke (former), ≥100 cigarettes in lifetime and currently smokes (current)), caloric intake (kcal/day), and urinary creatinine (g/L). Values reported are β (95% confidence interval).

**Table 4 nutrients-14-04271-t004:** Adjusted longitudinal associations ^a^ between quartiles of manganese and metabolic syndrome ^b^ in models with an interaction term between manganese and quartiles of other metals.

Interacting Metal Model	Mn Quartile 2 0.33–0.63 µg/L	Mn Quartile 3 0.63–1.13 µg/L	Mn Quartile 4 1.13–42.5 µg/L	*p-Value for Mn Trend*
Barium	1.44 (0.78, 2.68)	1.25 (0.57, 2.73)	1.00 (0.33, 3.08)	0.71
Cadmium	1.16 (0.64, 2.09)	1.14 (0.48, 2.71)	1.04 (0.39, 2.72)	0.80
Cobalt	1.56 (0.84, 2.90)	1.15 (0.53, 2.52)	1.99 (0.89, 4.45)	0.17
Cesium	1.80 (0.97, 3.35)	1.74 (0.74, 4.07)	1.05 (0.46, 2.41)	0.65
Molybdenum	1.69 (0.90, 3.17)	1.93 (0.89, 4.18)	1.22 (0.52, 2.88)	0.37
Lead	2.45 (1.27, 4.72)	1.97 (0.94, 4.10)	3.36 (1.11, 10.17)	0.01 *
Zinc	1.09 (0.59, 2.01)	1.50 (0.79, 2.83)	1.18 (0.41, 3.42)	0.37

* *p* < 0.05. ^a^ Associations between quartiles of manganese and metabolic syndrome incidence were assessed using Fine and Gray competing risks regression with the competing event of mortality. The reference group for manganese is the first quartile (≤0.33 µg/L). Values presented are sub-distribution hazard ratios (95% confidence interval) for each quartile of Mn. Models are adjusted for baseline values of the metal listed, a multiplicative interaction between Mn and the metal listed, sex, age (years), ethnicity (Hispanic, non-Hispanic), annual gross household income (<USD 10,000, USD 10,000–24,999, ≥USD 25,000), smoking status (<100 cigarettes in lifetime (never), ≥100 cigarettes in lifetime and does not currently smoke (former), ≥100 cigarettes in lifetime and currently smokes (current)), caloric intake (kcal/day), and urinary creatinine (g/L). ^b^ Metabolic syndrome was defined as having three or more of the following outcomes: (1) waist–hip ratio (waist circumference divided by iliac circumference) >0.90 for males, waist–hip ratio >0.85 for females, or a body mass index (measured weight divided by measured height squared) >30 kg/m^2^, (2) high density lipoprotein <40 mg/dL for males or <50 mg/dL for females, (3) triglycerides ≥150 mg/dL, (4) fasting glucose ≥100 mg/dL or diabetes (includes people who self-reported being diagnosed with diabetes via an oral glucose tolerance test or prescribed insulin or oral hyperglycemic medication), or (5) measured systolic blood pressure ≥130 mmHg or diastolic blood pressure ≥85 mmHg or self-reported current use of antihypertensive medications.

## Data Availability

Data sharing is not available for this study.

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
