# Peer review of "A Longitudinal Study of Exposure to Manganese and Incidence of Metabolic Syndrome"

_nutrients, 2022, doi:10.3390/nu14204271_

Round 1
Reviewer 1 Report
In this study, Riseberg et al assess longitudinal associations of Mn exposure with MetS and metabolic outcomes. They used data from the San Luis Valley Diabetes Study (SLVDS), a prospective cohort from rural Colorado with huge data collected from 1984-1998 (n=1,478). They assessed the shape of the cross-sectional association between Mn and MetS accounting for effect modification by other metals at baseline using Bayesian kernel machine regression. They also assessed longitudinal associations between baseline quartiles of Mn and incident MetS using Fine and Gray competing risks regression models and between quartiles of Mn and metabolic outcomes using linear mixed effects models. Quartiles of Mn were significantly associated with lower fasting glucose. Lead was found to be a possible effect modifier of the association between Mn and incident MetS. Authors concluded that Mn was associated with lower fasting glucose in this rural population and support a possible beneficial effect of Mn on diabetic markers. The authors use a large data set, which I believe has a certain significance. I have some questions.
1) Table 1 shows that those with higher Mn median (0.63 ug/L) were significantly more male, older, more hispanic, more never smoker, more obese, and less high fasting glucose. I may have missed it, but have these differences in background factors been corrected for, etc. in both the cross-sectional and longitudinal parts?
2) Line 596 or later is not required.
Reviewer 2 Report
In the article entitled “A longitudinal study of exposure to manganese and incidence 2 of metabolic syndrome by Riseberg et al presented the association of manganese (Mn) with metabolic syndrome (MetS) using a large number of clinical data set. Despite the study is very essential and contribute significantly to the field. There some aspects authors should address. Especially authors must give attention to the novelty of their studies and address issues related samples stability and measurement accuracy.
Abstract: Written well. Line 16-17, if possible, mention the nature of sample collected and analyzed for Mn i.e., human urine. Secondly, authors must indicate the concentration range of Mn detected in samples
Background: should be renamed as “introduction” and its well written.
Materials and method: Authors collected the samples between 1984-1988 and analyzed them now, its almost more than 30 years. What is the stability of Mn in urinary sample? Does it not oxidize? More details on sample measurements should be added. What’s the volume of urine sample used for analysis and how the concentrations were calculated.
Line 125-126: Authors must mention their method Mn detection limits.
Statistical analysis: mention the version and name of software used for correlation analysis etc.,
Table 1: What’s “All”? What does the number in brackets represents?
There is no term called “Low high-density lipoprotein”. Does authors measured LDL or HDL or its just a whole lipoprotein, mention it clearly.
Figure 2: the x-axes should be “Mn” not In
References are bit old. As it’s an interesting topic “latest literatures should be cited appropriately”.
Check thoroughly for English spells and grammar.
Round 2
Reviewer 2 Report
The authors addressed all the raised comments. As it's a clinical study with large data set, and the results are very promising, I strongly recommend the manuscript for publication.